# Highlighting a population-based re-emergence of Syphilis infection and assessing associated risk factors among pregnant women in Cameroon: Evidence from the 2009, 2012 and 2017 national sentinel surveillance surveys of HIV and syphilis

Cyprien Kengne-Nde[1]*, Jean de Dieu Anoubissi[1], Gabriel Loni-Ekali[1], Celine Nguefeu-Nkenfou[2,3], Yasmine Moussa[1], Arlette Messeh[1], Joseph Fokam[2,4,5], Albert Zeh-Meka[1], Denis Snayeul-Wawo[1], Dorine Tseuko[6], Marinette Ngo-Nemb[1], David Kob[7], Serge-Clotaire Billong[1,4,5], Leonard Bonono[1], Jean-Bosco Elat[1,5]

1 Central Technical Group, National AIDS Control Committee, Yaounde, Cameroon, 2 Chantal BIYA International Reference Centre (CBIRC) for research on HIV/AIDS prevention and management, Yaounde, Cameroon, 3 Higher Teachers' Training College, University of Yaounde I, Yaounde, Cameroon, 4 Faculty of Medicine and Biomedical Sciences (FMBS), University of Yaounde 1, Yaounde, Cameroon, 5 National HIV Drug Resistance Surveillance and Prevention Working Group, National AIDS Control Committee, Yaoundé, Cameroon, 6 National Laboratory of Public Health, Yaounde, Cameroon, 7 USAID, Yaounde, Cameroon

* cyprien.kengne@cnls.cm

## Abstract

### Background

Syphilis and HIV can be transmitted from pregnant women to their children and they remain a public health problem in Africa. Our study aimed to determine the trends of seroprevalence of HIV/syphilis co-infection and syphilis infection overtime through the national surveillance system in Cameroon and to explore associated risk factors.

### Methods

We conducted cross-sectional studies of HIV and syphilis, targeting each year 7000 first antenatal care (ANC-1) attendees at the same sites during the 2009, 2012 and 2017 sentinel surveillance surveys. Pregnant women were enrolled at their ANC-1, sociodemographic and clinical information were collected. HIV and Syphilis test were performed by serial algorithm as per the national guidelines. Trends were assessed for HIV, syphilis and HIV/syphilis by estimating seroprevalence from cross-sectional studies. Associated risk factors were explored using multinomial logistic regression with 4 outcomes: HIV/syphilis co-infection, HIV infection only, syphilis infection only and no infection.

### Results

Overall, 6 632, 6 521 and 6 859 pregnant women were enrolled in 2009, 2012 and 2017 respectively. In 2017, a total of 3 901 pregnant women enrolled were tested for syphilis.

**Data Availability Statement:** Data for 2009 and 2012 surveys cannot be made publicly available due to ethical restrictions. These data are available from the National Aids Control Committee Data Access (contact via infos@cnls.cm; Phone: (00237) 222 22 57 58, Fax: 222 23 34 39; P.O BOX: 1459 Yaounde, Cameroon). For 2017 data are available within the Supporting Information.

**Funding:** This study was funded by the Global Fund to fight AIDS, tuberculosis and malaria CMR_H_MOH_FM281019 (https://www.theglobalfund.org). The funders had no role in study design, data collection and analysis, decision to publish, or preparation of the manuscript.

**Competing interests:** The authors have declared that no competing interests exist.

**Abbreviations:** AIDS, Acquired Immune Deficiency Syndrome; ANC, Antenatal Care; aOR, adjusted Odds Ratio; ART, Antiretroviral Therapy; HIV, Human Immunodeficiency Virus; NLR, National Laboratory reference; PLHIV, People Living with HIV/AIDS; PMTCT, Prevention Mother To Child Transmission; STIs, Sexual Transmission Infections; WHO, World Health Organization.

Almost half of them (47.9%) were living in urban area and were aged less than 25 years (44.7%). While HIV epidemic was on a decline (from 7.6% (95% CI: 6.99–8.28) in 2009 to 5.7% (95% CI: 4.93–6.4) in 2017), a huge significant increase of syphilis prevalence was observed (from 0.6% (95% CI:0.40–0.80) in 2009 to 5.7% (95% CI:4.93–6.40) in 2017). Pregnant women residing in rural areas were more likely to be infected with syphilis than those living in the urban area (aOR = 1.8 [95% CI: 1.3–2.4]). Unmarried pregnant women were three time more likely to be infected by HIV/Syphilis Co-infection than married, cohabiting, widow or divorced pregnant women (aOR = 2.8 [95% CI: 1.3–2.4]). Furthermore; living in Northern region was associated with a lower risk of being infected with HIV (aOR = 0.6 [95% CI: 0.5–0.9]) and Syphilis infection (aOR = 0.6 [95% CI: 0.4–0.9]).

## Conclusion

The epidemiological dynamics of syphilis suggests a growing burden of syphilis infection in the general population of Cameroon. Our findings support the fact that while emphasizing strategies to fight HIV, huge efforts should also be made for strategies to prevent and fight syphilis infection especially among HIV positive women, in rural area, and southern regions.

## Introduction

HIV remains a public health problem especially in Sub-Saharan African countries. In 2018, 37.9 (32.7–44.0) million people globally were living with HIV while about 32 (23.6–43.8) million people have died from AIDS-related illnesses since the start of the epidemic [1]. About 540 000 [470 000–590 000] of them were living in Cameroon with AIDS-related deaths of 18 000 [15 000–21 000] people in the same year [2].

In Cameroon, the recent Population-based HIV Impact Assessment (CAMPHIA) conducted in 2017 reported an HIV prevalence of 3.7% in the population aged 15–64 years [3]. This prevalence varied by region, ranging from 6.3 percent in the South region to 1.5 percent in the Far North region, and by gender: females been twice more infected than males (5% and 2.3% respectively) [3].

The introduction and continuous availability of highly effective antiretroviral therapy (ART) have significantly reduced the mortality among people living with HIV/AIDS (PLHIV) worldwide and have transformed AIDS from an inevitably fatal condition to a chronic and manageable disease in some settings [4].

Sexual Transmitted Infections (STIs) such as HIV and Syphilis remain a public health problem in Africa and worldwide [9]. Many sub-populations like blood donors, armed forces, Men who have Sex with Men (MSM) are affected by these diseases [5–10]. HIV and Syphilis are two of common pathogens to which many pregnant women in Sub-Saharan African Countries are exposed or infected [5–11]. They can be transmitted vertically during pregnancy to their children. Pregnant women diagnosed with early stages of syphilis are most at risk to transmit syphilis to their infants, 60% to 90% of pregnant women with untreated primary or secondary syphilis will transmit syphilis to their fetus as compared to less than 10% of women with late latent syphilis [12]. Access to screening during pregnancy has improved over the past decade in Sub-Saharan African Countries generally and in Cameroon particularly as HIV prevention programs have been scaled up. The Screening of such infections during antenatal care visit is important since many of these infections are asymptomatic [9].

Moreover, syphilis is an important risk factor in HIV transmission. Syphilis has been implicated in susceptibility to Human Immunodeficiency Virus (HIV) infection with an odds ratio of 8.5 for men and 3.3 for women [13]. If the infection remains undiagnosed, complications of late stage syphilis will develop in a portion of affected individuals [11].

Syphilis and HIV continue to be significant problems in Cameroon. Hence, identifying the risk factors and analyzing the trends would serve as a footprint to set-up evidence based interventions to reduce the burden of HIV and Syphilis among pregnant women and would benefit global efforts to eliminate both diseases in infants [14–16]. We therefore sought to describe the prevalence trends of syphilis infection and HIV/Syphilis co-infection among pregnant women using sentinel surveillance surveys data over a nine-year period and to investigate the associated risk factors.

## Methods

### Description of the surveys, study design and participants

To determine trends of syphilis prevalence over nine-year period, data from consecutive ANC cross sectional analytic surveys conducted in 2009, 2012 and 2017 were examined. As recommended by WHO for HIV sentinel surveillance survey [17], from each year, a cross sectional analytic study was conducted in 20 sentinels survey sites across the 10 Regions of Cameroon, which included 60 HIV surveillance health facilities (routine collection points). A sentinel site is a specific geographic area in the region. In the sentinel site, HIV surveillance health facilities (clinic and or hospital) were the data collection points. Each sentinel site had 3 data collection points. We have implemented a non-probabilistic approach which consisted of a systematic sampling method with two stages: (1) Selection of sentinel sites and surveillance health facilities (routine collection points), (2) Selection of pregnant women in each study sentinel site. The health facilities were chosen based on their capacity to provide both ANC and PMTCT services, their location (urban and rural settings in each region of the country), ANC attendance (capacity to enroll at least 300 pregnant ANC1 attendees during the study period of three months). At each site, pregnant women aged 15–49 years attending their first ANC were enrolled consecutively until the sample size was reached. The sample size for each region was determined as recommended by WHO guidelines [17] taking into considerations the observed prevalence of HIV and a desired precision of 95%. The minimum number of pregnant women at first ANC required to estimate the prevalence of HIV infection in that population with precision $i = 0.03$ and for a fixed risk $\alpha = 0.05$ ($Z_{\alpha/2} = 1.96$) is given by the following formula:

$$n = \frac{Z^2_{\alpha/2}}{i^2} P(1 - P)$$

where $P = $ = observed prevalence

$i$ = precision of the estimation

$Z_{\alpha/2}$ = Quantile for a risk $\alpha = 0,05$

### HIV and syphilis testing procedure

**HIV testing.** Plasma samples and sociodemographic data were collected as per the routine clinical practice per site. During the study period, HIV screening was proposed to every ANC-1 attendee according to the serial algorithm recommended by the Ministry of Public Health in Cameroon [18]. After onsite HIV testing, residual plasmas were transferred into cryotubes labelled with a specific ID code of the participant, and stored at 0°-8°C. Plasma and accompanying sample sheet were then linked by the ID code. Labeled samples were shipped at the National Reference Laboratory (NRL) following universal standards for transport of plasma specimens [19].

A serial algorithm for HIV screening was used in all PMTCT-site laboratories and at the NRL, following the national algorithm for voluntary HIV testing (Fig 1A). Briefly, a sample was assayed by the first test Determine HIV1/2 (*Abbott*, *Minato-ku-Tokyo*, *Japan*); non-reactive samples were reported as negative while in case of reactivity to test-1, a second test was then used for confirmatory analysis (Oraquick). Onsite, indeterminate HIV results were reported as indeterminate, while specimens with indeterminate/discordant HIV results at the NRL were tested with ELISA (ImmunoComb II HIV 1&2 BiSpot), used as tiebreaker. Residual plasma was stored at -70°C at the NRL for quality control or further testing if necessary.

**Syphilis testing.**   Syphilis testing was performed in all PMTCT-site laboratories using the Treponema Palladium Hemagglutination assay (TPHA) / Veneral Diseases Research Laboratory (VDRL) as per the manufacturer's instructions and WHO recommendations (Fig 1B) [20].

## Data collection

Socio-demographic and clinical characteristics were collected from consenting pregnant ANC1 attendees by a nurse without altering the normal functional routine of the health facility. After completing the questionnaire, participants were sent to the laboratory where plasma was collected for HIV and syphilis screening tests according to the routine procedure at the site. There were three focal persons per sentinel site, consisting of the director of the health facility, an Antenatal Care (ANC) staff, and a laboratory technician. All received four-day training on the protocol, laboratory techniques related to the surveillance, standard operating procedures of data collection and blood sample collection. For all consecutive surveys, data were collected using a surveillance questionnaire for all eligible and consenting pregnant women and socio-demographic information were collected by trained ANC-PMTCT staff. The questionnaires were maintained at the sentinel sites and transported to the research unit of the National AIDS Control Committee (NACC) and the plasma samples were sent to the NRL for analyses.

## Data management and statistical analysis

Data was entered in a computer using the CSPRO version 6.2.0 (U.S Census Bureau, ICF International, and Serpro S.A., Washington, DC 20233–8860, USA). The analyses were performed using STATA/SE version 13.0 (STATACORP, Texas, USA). Pregnant women with

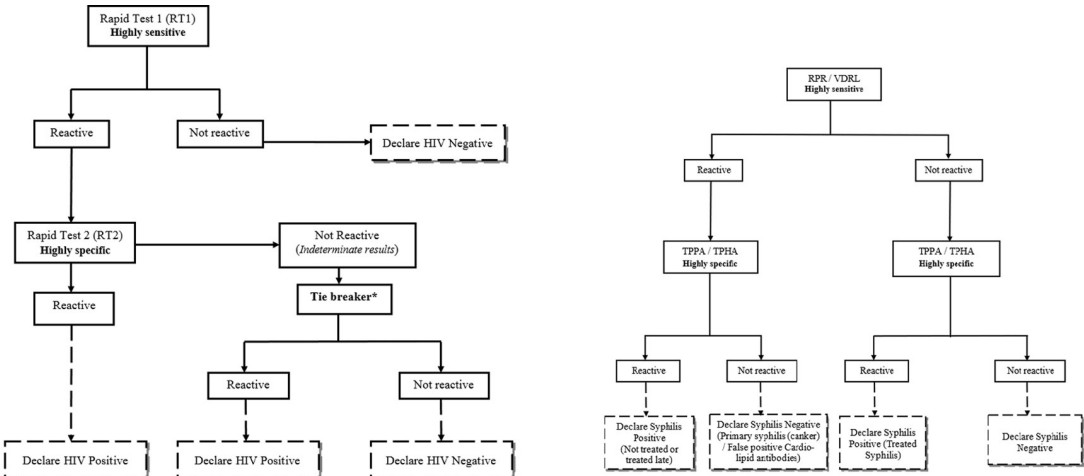

**Fig 1.  a.** Algorithm for HIV rapid testing during a voluntary and free counseling. **b.** Algorithm for syphilis rapid testing during a voluntary and free counseling.

inconclusive test results were not considered in the analysis. Those who could not benefit from syphilis testing due to stock out were also excluded from the analysis. Continuous variables were reported by median with Interquartile interval range (25th and 75th percentiles) and by mean and standard deviation, while categorical variables were described as frequencies and percentages. The overall prevalence of HIV/syphilis co-infection, Syphilis infection and HIV infection among pregnant women was estimated with an exact binomial confidence intervals (CI) at 95%. Only data from the 2017 (a subset where women were tested for both HIV in the NRL and Syphilis in PMTCT sites) were used to assessed factors associated with HIV/syphilis co-infection. The main outcome (dependent variable) of our analysis was HIV/syphilis co-infection variable which has 4 categories: category 1 = HIV/syphilis co-infection; category 2 = HIV infection only; category 3 = syphilis infection only; category 4 = no infection. A multi-variate multinomial logistic regression was used to investigate the factors associated with HIV/syphilis co-infection among pregnant ANC-1 attendees.

Factors that were associated with HIV/syphilis co-infection in the univariate analysis at 0.25 significance level were considered when developing the final multivariate model. To iden-tify potential risk factors, a multivariate analysis using a backward stepwise elimination method was used. All variables that were significant at 5% in multivariate analysis were kept into the final model. The risk factors for syphilis infection were determined using co-variates fitted in the multi-variable logistic regression analysis: age, HIV infection, marital status, edu-cation level, occupation, residence, and parity. A likelihood ratio test was used to check for interactions and in determining the best multi-variable model for the data.

### Ethical considerations

Ethical clearance for the surveys was obtained from the Cameroon National Ethics Commit-tee (ref N˚2017/03/879/CECNERSH/SP). The privacy of consenting pregnant women and data confidentiality were ensured by the use of ID codes. All participants had signed informed consent without any incentive. HIV tests were offered for free and all women tested positive were placed on ART according to the national guidelines. Confidentiality and privacy of the study subjects was ensured by permanently delinking personal identifiers with subject information.

## Results

### Description of population in 2009, 2012 and 2017 in Cameroon

Over the 7000 pregnant women targeted, the participation rate was higher in 2017 (2017: 98.0%) and was above 90% in the two other surveys (2009: 94.7%, 2012: 93.2%) (Table 1). Overall, the pregnant women enrolled across the threee surveys were relatively young. In fact, 53% of them were aged between 15–24 years in 2009, almost half (49.3% were aged between 15–24 years) in 2012 and 42.7% were aged between 15–24 years in 2017 (Table 1).The median of age sligthly increase overtime going from 24 years (IQR [20–28]) in 2009 to 26 years [21–30] in 2017.

Morever half or more of the women enrolled were housewive (57% in 2009; 49.7% in 2012 and 49.4% in 2017) and the majority of them were married or cohabiting (80.3% in 2009 and 77.4% in 2017). Unfortunaly this variable was missed in the 2012 survey. In addition, the pro-portion of pregnant women who reached university sligthly increased over the year (6% in 2009; 8.3% in 2012 and 13.5% in 2017) and more than half of them were living in urban area in all the three surveys (60% in 2009; 57.3% in 2012 and 57.3% in 2017) (Table 1).

**Table 1. Demographics characteristics among pregnant women aged beyond 15 years old, national sentinel surveillance survey of HIV and syphilis in 2009 (n = 6632), 2012 (n = 6521) and 2017 (n = 6859).**

| Variable | 2009 (n = 6632) | | 2012 (6521) | | 2017 (n = 6859) | |
|---|---|---|---|---|---|---|
| | Number | Percentage | Number | Percentage | Number | Percentage |
| Age (Median [IQR]) | 24 [20–28] | | 25 [21–29] | | 26 [21–30] | |
| **Age** | | | | | | |
| 15–19 | 1525 | 23.0 | 1193 | 18.3 | 1033 | 15.1 |
| 20–24 | 1990 | 30.0 | 2024 | 31.0 | 1896 | 27.6 |
| 25–29 | 1592 | 24.0 | 1676 | 25.7 | 1886 | 27.5 |
| 30–34 | 928 | 14.0 | 1041 | 16.0 | 1282 | 18.7 |
| 35–49 | 597 | 9.0 | 587 | 9.0 | 762 | 11.1 |
| **Level of education** | | | | | | |
| None | 1260 | 19.0 | 1071 | 16.4 | 949 | 13.8 |
| Primary | 2122 | 32.0 | 2319 | 35.6 | 1712 | 24.9 |
| Secondary | 2852 | 43.0 | 2592 | 39.8 | 3275 | 47.8 |
| University | 398 | 6.0 | 539 | 8.3 | 923 | 13.5 |
| **Occupation** | | | | | | |
| Housewife | 3774 | 56.9 | 3241 | 49.7 | 3385 | 49.4 |
| Student | 889 | 13.4 | 926 | 14.2 | 1168 | 17 |
| Employees | 1486 | 22.4 | 2152 | 33.0 | 1604 | 23.4 |
| Other | 484 | 7.3 | 202 | 3.1 | 702 | 10.2 |
| **Marital status** | | | | | | |
| Single | 1240 | 18.7 | NA | NA | 1504 | 21.9 |
| Married / Cohabiting | 5326 | 80.3 | NA | NA | 5305 | 77.4 |
| Widow/ Divorced | 67 | 1.0 | NA | NA | 50 | 0.7 |
| **Area of residence** | | | | | | |
| Urban | 3979 | 60 | 2787 | 57.3 | 3932 | 57.3 |
| Rural | 2653 | 40 | 3734 | 42.7 | 2927 | 42.7 |

## Evolution of Syphilis HIV and co-infection among pregnant women from 2009 to 2017

The overall prevalence of syphilis increased hugely going from 0.6% (95% CI:0.40–0.80) in 2009 to 5.6% (95% CI:4.93–6.40) in 2017 (Fig 2). This increasing trend was also observed for the coinfection HIV/syphilis ranging from 0.05% (95% CI:0.01–0.13) in 2009 to 0.49% (95% CI:0.29–0.76) in 2017. This prevalence is five times higher in 2012 than in 2009 (2.93% vs. 0.6%, p <0.001) and almost nine times higher in 2017 than in 2009 (5.6% vs 0.6%, p<0.001). However, we found a significant decrease in the prevalence of HIV infection from 7.6% (95% CI: 6.99–8.28) in 2009 to 5.7% (95% CI: 4.93–6.4) in 2017.

## Description of population and distribution of syphilis and HIV/syphilis co-infection in 2017 in Cameroon

In 2017, a total of 3901 pregnant women (56.9% of the total pregnant women enrolled) were tested for syphilis. Almost half of them (47.9%) were living in urban area and were aged less than 25 years (44.7%). About three fifth (62.9%) attended at least the secondary education level. About three fourth (75.1%) were married or cohabiting, and over two fifths (43.4%) were housewife and 23.9% were nulliparous (Table 2).

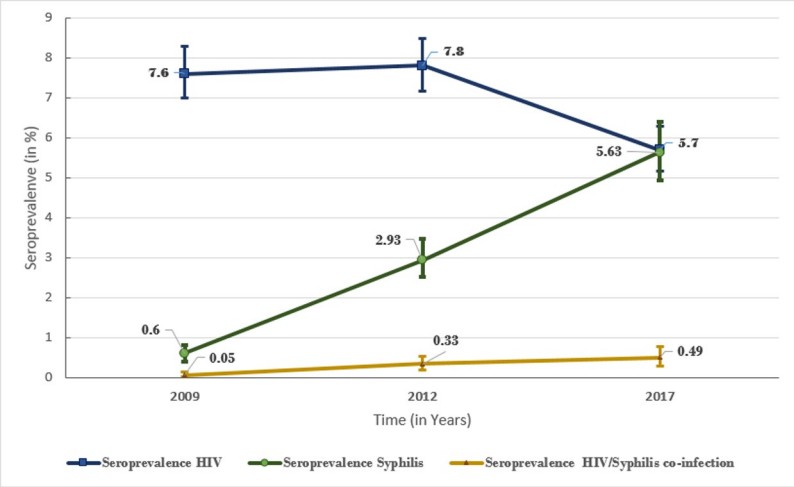

Error bars represented 95% Exact Confidence Intervals

**Fig 2. Evolution of seroprevalence of HIV/syphilis co-infection, HIV and syphilis from 2009 to 2017 in Cameroon.**

Among the pregnant women who were tested for syphilis, a total of 222 HIV tests results were positive for HIV giving a sero-prevalence of 5.8% of HIV infection, and 24% of them were aged less than 25 years old (Table 2).

According to HIV status, prevalence of syphilis was higher among HIV positive pregnant women than HIV negative (8.6% vs 5.3%, p = 0.049), with a borderline significance (Table 3). Moreover, the prevalence of syphilis was higher in rural area than in urban area (7.2% vs 4.2%, p < 0.0001) and in Southern Regions (Centre, East, Littoral, North West, West and South) than Northern (Adamawa, North and Far North) one (6.2 vs 4.0, p = 0.005), there was no significant difference among HIV + population (8.5% vs 8.7%, p = 1; 8.5 vs 8.7, p = 0.611) according to residence and regions respectively.

Among pregnant HIV positive women, we found that prevalence of syphilis was higher among nulliparous women than non-nulliparous women (20.8% vs 7.1%, p = 0.039).

## Factor associated with Syphilis and HIV/syphilis co-infection in 2017 in Cameroon

The multivariate multinomial logistic regression revealed that residence localization was associated with syphilis infection. After adjusting for the effects of other variables, pregnant women residing in rural areas were more likely to be infected with syphilis than those living in the urban area (aOR = 1.8 [95% CI: 1.3–2.4]) (Table 4). We also found that age and type of union were associated with HIV/syphilis co-infection. In fact, single or unmarried pregnant women were three times more likely to be co-infected by HIV and syphilis than married, cohabiting, widow or divorced pregnant women (aOR = 2.9 [95% CI: 1.0–8.2]). Pregnant women aged 25–49 years were strongly more likely to be co-infected (aOR = 15.1 [95% CI: 3.0–75.7]) by HIV and syphilis than those aged 15–24 years. Furthermore, region; type of union, nulliparous and age were associated to HIV infection. Living in Northern Region (Adamawa, North and Far North) was associated with a lower risk of being infected with HIV (aOR = 0.6 [95% CI: 0.5–0.9]) and syphilis infection (aOR = 0.6 [95% CI: 0.4–0.9]) than living in Southern Regions (Centre, East, Littoral, North West, West and South), but being single or unmarried was associated with a higher risk of HIV infection (aOR = 1.5 [95% CI: 1.1–2.1]) than being in couple (married or not), divorced or widow. In the same way, having 25–49

**Table 2.** Demographics characteristics among pregnant women aged beyond 15 years old who were tested for HIV and syphilis, national sentinel surveillance survey of HIV and syphilis 2017.

| Characteristic | Overall Population (N = 3901) | | Among HIV+ Population (N = 222) | |
| --- | --- | --- | --- | --- |
| | Number | Percentage | Number | Percentage |
| **HIV Status** | | | | |
| HIV positive | 222 | 5.8 | NA | NA |
| HIV negative | 3,574 | 94.2 | NA | NA |
| **Total** | **3796** | **100.0** | **NA** | **NA** |
| **Residence** | | | | |
| Urban | 2,031 | 52.1 | 118 | 53.2 |
| Rural | 1,870 | 47.9 | 104 | 46.8 |
| **Total** | **3901** | **100.0** | **222** | **100.0** |
| **Region** | | | | |
| Adamawa | 477 | 12.2 | 15 | 6.8 |
| Centre | 559 | 14.3 | 49 | 22.1 |
| East | 560 | 14.4 | 46 | 20.7 |
| Far North | 365 | 9.4 | 8 | 3.6 |
| Littoral | 76 | 2.0 | 4 | 1.8 |
| North | 116 | 3.0 | 0 | 0.0 |
| North West | 211 | 5.4 | 11 | 5.0 |
| West | 240 | 6.2 | 6 | 2.7 |
| South | 614 | 15.7 | 28 | 12.6 |
| South West | 683 | 17.5 | 55 | 24.8 |
| **Total** | **3901** | **100.0** | **222** | **100.0** |
| **Type of union** | | | | |
| Single | 939 | 24.6 | 64 | 29.5 |
| Married/Cohabiting | 2,862 | 75.1 | 151 | 69.6 |
| Widow/Divorced | 12 | 0.3 | 2 | 0.9 |
| **Total** | **3813** | **100.0** | **217** | **100.0** |
| **Education** | | | | |
| None | 452 | 11.7 | 18 | 8.2 |
| Primary | 981 | 25.4 | 71 | 32.4 |
| Secondary | 1,929 | 49.9 | 25 | 11.4 |
| Higher | 504 | 13.0 | 105 | 48.0 |
| **Total** | **3866** | **100.0** | **219** | **100.0** |
| **Primiparous** | | | | |
| Yes | 931 | 23.9 | 24 | 10.8 |
| No | 2,956 | 76.1 | 198 | 89.2 |
| **Total** | **3887** | **100.0** | **222** | **100.0** |
| **Number of pregnancies** | | | | |
| 0 | 931 | 23.9 | 24 | 10.9 |
| 2 to 3 | 1,496 | 38.5 | 78 | 35.1 |
| 4 to 5 | 888 | 22.9 | 78 | 35.1 |
| >5 | 572 | 14.7 | 42 | 18.9 |
| **Total** | **3887** | **100.0** | **222** | **100.0** |
| **Age (in years)** | | | | |
| **<25** | **1,734** | **44.7** | **53** | **24.0** |
| 15–19 | 665 | 17.1 | 7 | 3.2 |
| 20–24 | 1,069 | 27.6 | 46 | 20.8 |

*(Continued)*

**Table 2.** (Continued)

| Characteristic | Overall Population (N = 3901) | | Among HIV+ Population (N = 222) | |
|---|---|---|---|---|
| | Number | Percentage | Number | Percentage |
| ≥ 25 | 2,144 | 55.3 | 168 | 76.0 |
| 25–29 | 1,011 | 26.1 | 63 | 28.5 |
| 30–34 | 721 | 18.6 | 59 | 26.7 |
| 35–49 | 412 | 10.6 | 46 | 20.8 |
| Total | 3878 | 100.0 | 221 | 100.0 |
| Occupation | | | | |
| Housewife | 1,655 | 43.4 | 91 | 42.3 |
| Student | 727 | 19.0 | 19 | 8.8 |
| Formal employee (Civil servant) | 308 | 8.1 | 17 | 7.9 |
| Informal employee/entrepreneurship | 889 | 23.3 | 72 | 33.5 |
| Unemployed | 163 | 4.3 | 4 | 1.9 |
| Others | 74 | 1.9 | 12 | 5.6 |
| Total | 3816 | 100.0 | 215 | 100.0 |

NA: Not Applicable

years old or being multiparous were associated with a higher risk of HIV infection (aOR = 2.7 [95% CI: 1.7–4.3]; aOR = 2.0 [95% CI: 1.4–2.6] respectively).

## Discussion

The aim of our study was to describe the prevalence trends of syphilis and HIV/syphilis co-infection among pregnant women using sentinel surveillance surveys over a nine-year period and investigate the risk factors associated to HIV co-infection.

To date, maternal syphilis seroprevalence maybe increasing in Cameroon. From 2009 to 2017, the prevalence of syphilis among pregnant women attending ANC in Cameroon increased significantly from 0.6% (95% CI:0.40–0.80) in 2009 to 5.6% (95% CI:4.93–6.40). HIV and syphilis co-infection is also increasing, due to the re-emergence of syphilis among ANC-1 attendees in Cameroon. The prevalence of HIV/syphilis co-infection significatively increased also from 0.05% (95% CI:0.01–0.13) in 2009 to 0.49% (95% CI:0.29–0.76) in 2017. Although the prevalence of HIV/syphilis co-infection is still low (<1%), it was ten times higher in 2017 than in 2009 (0.49% vs 0.05%, p<0.001). In Sub-Saharan African countries, the prevalence of syphilis among pregnant women was estimated to range from 0.6% in Senegal to 14.0% in Equatorial Guinea [21–24]. In 2017, the prevalence of syphilis found among pregnant women attending antenatal clinics in Cameroon was among the middle's ones in Sub-Saharan African countries (5.6%). The prevalence of syphilis among pregnant women in Cameroon was higher in rural than in urban areas. The same result was found in Tanzania where syphilis prevalence was highest among pregnant women from remote rural residents (16.0%) compared to urban residents (7.0%) [25]. In contrast, a study conducted in Zambia found the opposite and established that syphilis prevalence was highest among pregnant women residing in urban residence (9.2%) as compared to those residing in rural (7.8%) [27].

Moreover, the prevalence of syphilis was higher among HIV positive pregnant women than HIV negative (8.6% vs 5.3%). Similar result has also been found in Rwanda in 2011, where syphilis prevalence was highest among HIV-positive pregnant women (10.8%) compared to HIV negative pregnant women (1.8%) [9]. This could due to the fact that pregnant women

**Table 3. Prevalence of Syphilis among pregnant women aged beyond 15 years old who were tested for HIV and syphilis by selected demographic characteristics, national sentinel surveillance survey of HIV and syphilis 2017.**

| Characteristic | Overall Population (N = 3901) | | | | | | p-value# | Among HIV+ Population (N = 222) | | | | | | p-value# |
|---|---|---|---|---|---|---|---|---|---|---|---|---|---|---|
| | Syphilis + | | Syphilis - | | Total | | | Syphilis + | | Syphilis - | | Total | | |
| | n | % | n | % | n | % | | n | % | n | % | n | % | |
| **HIV Status**[b] | | | | | | | **0.049** | | | | | | | NA |
| HIV positive | 19 | 8.6 | 203 | 91.4 | **222** | **100.0** | | NA | NA | NA | NA | **NA** | **NA** | |
| HIV negative | 191 | 5.3 | 3,383 | 94.7 | **3574** | **100.0** | | NA | NA | NA | NA | **NA** | **NA** | |
| **Residence** | | | | | | | **<0.0001** | | | | | | | 1 |
| Urban | 85 | 4.2 | 1,946 | 95.8 | **2031** | **100.0** | | 10 | 8.5 | 108 | 91.5 | **118** | **100.0** | |
| Rural | 135 | 7.2 | 1,735 | 92.8 | **1870** | **100.0** | | 9 | 8.7 | 95 | 91.4 | **104** | **100.0** | |
| **Region Group** | | | | | | | **0.005** | | | | | | | 0.611 |
| Northern | 38 | 4.0 | 920 | 96.0 | **958** | **100.0** | | 2 | 8.7 | 21 | 91.3 | **23** | **100.0** | |
| Southern | 182 | 6.2 | 2,761 | 93.8 | **2943** | **100.0** | | 17 | 8.5 | 182 | 91.5 | **199** | **100.0** | |
| **Type of union** | | | | | | | 0.372 | | | | | | | 0.178 |
| Single | 59 | 6.3 | 880 | 93.7 | **939** | **100.0** | | 8 | 12.5 | 56 | 87.5 | **64** | **100.0** | |
| Married/Cohabiting/ Widow/Divorced | 158 | 5.5 | 2,716 | 94.5 | **2874** | **100.0** | | 10 | 6.5 | 143 | 93.5 | **153** | **100.0** | |
| **Education** | | | | | | | **0.019** | | | | | | | 0.112 |
| None | 22 | 4.9 | 430 | 95.1 | **452** | **100.0** | | 3 | 16.7 | 15 | 83.3 | **18** | **100.0** | |
| Primary | 74 | 7.5 | 907 | 92.5 | **981** | **100.0** | | 4 | 5.6 | 67 | 94.4 | **71** | **100.0** | |
| Secondary | 101 | 5.2 | 1,828 | 94.8 | **1929** | **100.0** | | 6 | 5.7 | 99 | 94.3 | **105** | **100.0** | |
| Higher | 20 | 4.0 | 484 | 96.0 | **504** | **100.0** | | 4 | 16.0 | 21 | 84.0 | **25** | **100.0** | |
| **Primiparous** | | | | | | | 0.172 | | | | | | | **0.039** |
| Yes | 59 | 6.3 | 872 | 93.7 | **931** | **100.0** | | 5 | 20.8 | 19 | 79.2 | **24** | **100.0** | |
| No | 161 | 5.5 | 2795 | 94.6 | **2956** | **100.0** | | 14 | 7.1 | 184 | 92.9 | **198** | **100.0** | |
| **Number of pregnancies** | | | | | | | 0.128 | | | | | | | **0.005** |
| 0 | 59 | 6.3 | 872.0 | 93.7 | **931** | **100.0** | | 5 | 26.3 | 19 | 9.4 | **24** | **100.0** | |
| 2 to 3 | 71 | 4.8 | 1,425 | 95.3 | **1555** | **100.0** | | 1 | 5.3 | 77 | 37.9 | **78** | **100.0** | |
| 4 to 5 | 49 | 5.5 | 839 | 94.5 | **888** | **100.0** | | 8 | 42.1 | 70 | 34.5 | **78** | **100.0** | |
| >5 | 41 | 7.2 | 531 | 92.8 | **572** | **100.0** | | 5 | 26.3 | 37 | 18.2 | **42** | **100.0** | |
| **Age (in years)** | | | | | | | 0.199 | | | | | | | 0.258 |
| 15–24 | 104 | 6.0 | 1,630 | 94.0 | **1734** | **100.0** | | 2 | 3.8 | 51.0 | 96.2 | **53** | **100.0** | |
| 25–49 | 114 | 5.3 | 2,030 | 94.7 | **2144** | **100.0** | | 17 | 10.1 | 151 | 89.9 | **168** | **100.0** | |
| **Occupation** | | | | | | | 0.297 | | | | | | | 0.982 |
| Housewife | 83 | 5.0 | 1572 | 95.0 | **1655** | **100.0** | | 8 | 8.8 | 83 | 91.2 | **91** | **100.0** | |
| Student | 44 | 6.1 | 683 | 94.0 | **727** | **100.0** | | 2 | 10.5 | 17 | 89.5 | **19** | **100.0** | |
| Formal employee (Civil servant) | 13 | 4.2 | 295 | 95.8 | **308** | **100.0** | | 1 | 5.9 | 16 | 94.1 | **17** | **100.0** | |
| Informal employee/entrepreneurship | 59 | 6.6 | 830 | 93.4 | **889** | **100.0** | | 6 | 8.3 | 66 | 91.7 | **72** | **100.0** | |
| Unemployed & Others | 16 | 6.8 | 221 | 93.3 | **237** | **100.0** | | 0 | 6.3 | 15 | 93.8 | **15** | **100.0** | |

#: Fisher's Exact Test

[b]Not Applicable (NA) in the HIV-syphilis co-infection cross table.

living with HIV may have a weaker immune system than other pregnant women. Preventives strategies should prioritize this group of the population.

In contrast of syphilis trend, our study found a significantly decrease of HIV infection among pregnant women from a prevalence of 7.6% (95% CI: 6.99–8.28) in 2009 to 5.7% (95% CI: 4.93–6.4) in 2017. We observe the same trend by triangulate with DHIS data collected routinely among pregnant women in the country. This trend may be the result of several strategies

**Table 4. Risk factors associated to HIV/syphilis co-infection among pregnant women aged beyond 15 years old who were tested for HIV and syphilis, national sentinel surveillance survey of HIV and syphilis 2017.**

| Characteristics | Univariate Analysis | | | Multivariate Analysis | | |
|---|---|---|---|---|---|---|
| | HIV/syphilis Co-infection vs No infection | HIV infection vs No infection | Syphilis infection vs No infection | HIV/syphilis Co-infection vs No infection | HIV infection vs No infection | Syphilis infection vs No infection |
| | OR [95% IC] | OR [95% IC] | OR [95% IC] | aOR [95% IC] | aOR [95% IC] | aOR [95% IC] |
| **Residence** | | | | | | |
| Urban | Ref | Ref | Ref* | Ref | Ref | Ref |
| Rural | 1 [0.4–2.5 ] | 0.8 [0.7–1.1 ] | 1.9 [1.4–2.5 ] | 1.1 [0.4–2.8 ] | 0.9 [0.7–1.1 ] | 1.8 [1.3–2.4 ] [μ] |
| **Region Group** | | | | | | |
| Northern | 0.3 [0.1–1.4 ] | 0.6 [0.4–0.8 ] | 0.6 [0.4–0.9 ] | 0.5 [0.1–2.1 ] | 0.6 [0.5–0.9 ] [μ] | 0.6 [0.4–0.9 ] [μ] |
| Southern | Ref | Ref | Ref* | Ref | Ref | Ref |
| **Type of union** | | | | | | |
| Single | 2.5 [1–6.4 ] | 1.1 [0.8–1.4 ] | 1.1 [0.8–1.5 ] | 2.9 [1–8.2 ] [a] | 1.5 [1.1–2.1 ] [μ] | 1 [0.7–1.5 ] |
| Married/Cohabiting/ Widow/Divorced | Ref | Ref | Ref* | Ref | Ref | Ref |
| **Education** | | | | | | |
| None | Ref | Ref | Ref* | | | |
| Primary | 0.7 [0.1–3 ] | 2.1 [1.3–3.3 ] | 1.8 [1.1–3.1 ] | | | |
| Secondary | 0.5 [0.1–1.9 ] | 1.4 [0.9–2.3 ] | 1.2 [0.7–2 ] | | | |
| Higher | 1.2 [0.3–5.4 ] | 1.1 [0.6–2 ] | 0.8 [0.4–1.5 ] | | | |
| **Primiparous** | | | | | | |
| Yes | Ref | Ref | Ref* | Ref | Ref | Ref |
| No | 0.9 [0.3–2.6 ] | 3 [2.1–4.5 ] | 0.9 [0.6–1.2 ] | 0.4 [0.1–1.2 ] | 2.7 [1.7–4.3 ] [μ] | 1 [0.7–1.4 ] |
| **Number of pregnancies** | | | | | | |
| 0 | Ref | Ref | Ref* | | | |
| 2 to 3 | 0.1 [0–1.1 ] | 2.6 [1.7–4 ] | 0.8 [0.6–1.2 ] | | | |
| 4 to 5 | 1.8 [0.6–5.5 ] | 3.8 [2.5–5.8 ] | 0.9 [0.6–1.3 ] | | | |
| >5 | 1.7 [0.5–6 ] | 3.1 [1.9–4.9 ] | 1.2 [0.7–1.8 ] | | | |
| **Age (in years)** | | | | | | |
| 15–24 | Ref | Ref | Ref* | Ref | Ref | Ref* |
| 25–49 | 7.2 [1.7–31.2 ] | 2.4 [1.9–3.2 ] | 0.8 [0.6–1.1 ] | 15.1 [3–75.7 ] [μ] | 2 [1.4–2.6 ] [μ] | 0.8 [0.6–1.2 ] |
| **Occupation** | | | | | | |
| Housewife | Ref | Ref | Ref* | | | |
| Student | 0.6 [0.1–2.6 ] | 0.4 [0.3–0.6 ] | 1.3 [0.8–1.8 ] | | | |
| Formal employee (Civil servant) | 0.7 [0.1–5.4 ] | 1.2 [0.7–1.8 ] | 0.9 [0.5–1.6 ] | | | |
| Informal employee/ entrepreneurship | 1.5 [0.5–4.3 ] | 1.6 [1.2–2.2 ] | 1.4 [1 – 2 ] | | | |
| Unemployed & Others | 0.9 [0.1–7.4 ] | 1.3 [0.8–2.1 ] | 1.5 [0.8–2.6 ] | | | |

[a]: attempted a borderline.

Ref: reference category

#: Global p-value < 0.25

*: Global p-value < 0.05

[μ]: P-value < 0.05

implemented by the country to strengthen the prevention of HIV particularly among pregnant women [28–30].

Living in a rural area was a risk factor for syphilis infection. This can be explaining by the fact that some pregnant women do not attend ANC services early and some of the rural area facilities may not have syphilis test permanently for screening pregnant women. Furthermore, in this area people may not use condom frequently and may not aware of its importance. As recommended in the WHO guide, syphilis (new or old infection) is cure in first intention with benzathine penicillin [20]. Thus the global stockout of benzathine penicillin in 2016 could have play a role in the sero-prevalence rate and the spread of syphilis, particularly in rural areas [30]. We also found that living in Northern regions (Adamawa, North and Far North) was associated with a lower risk for both syphilis infection as well as HIV infection. Southern regions have the most affected regions which include South, East, Centre and North-West. It is worthy to note that Northern regions are more muslim and southern ones are more Christian. These regions should therefore be considered for priority interventions to curb this high HIV and syphilis burden.

As found in other studies [9, 27], being single and being aged beyond 25 years old were associated with a higher risk of HIV/syphilis co-infection and HIV infection. Possible reasons that may explain these results include that single women may have several sexual partners before moving in a stable relationship and due to their economic vulnerability, they are more involved in transactional sex which increases their exposure to HIV and syphilis as a result [31–33]. Women aged beyond 25 years old and multiparous were more likely to be exposed to HIV and syphilis longer than younger, since they were sexually active.

The strength of our study is the representative sample size which covered the whole Regions of the country and offers the possibility to generalize our results [27].

This study has some limitations. Several sexual risk factors were not collected in these surveys and therefore more data are needed on sexual behaviors and sexual partners experience. Data were pooled across all sites for analysis, and therefore site-level variation was not accounted for in our analysis. This may have impacted our trends assessment but we believe that these limitations did not significantly affect the final interpretation of the overall trend of our study findings. The funding allocated to the project was not enough to provide syphilis tests at the study sites, neither at the NRL. Then syphilis test was done in sentinel sites which had the test for their routine ANC services during study period. Therefore, all pregnant women enrolled in our surveys were not tested for syphilis due to stock out at the level of some sentinel sites. This might have slightly affected the power of our analyses by reducing the sample size of analyses, and some participants who might be positive for syphilis infection were not included in this analysis. In addition, information on ART for the women who were tested positive to HIV were not collected. This could be used to improve future Sentinel Surveillance Surveys and the WHO protocol on this aspect in the area of Test and Treat. Moreover, the titres were measured for VDRL for the majority of the sentinel sites to determine if it was a current infection or an old or scarring infection, but this information was not collected during the survey. This could have helped to further refine our factor analysis of the re-emergence of syphilis cases in the country.

## Conclusion

Antenatal care provides an excellent opportunity to screen women for infections which are common and treatable and can be transmitted vertically. HIV and syphilis infections remain common in pregnant women. Sentinel surveillance surveys provide a good opportunity to collect data which could provide useful information for decision making, especially in the era of dual elimination (HIV and syphilis). The data of these surveys show that syphilis is becoming a major public health problem in Cameroon. The trends of syphilis among pregnant women

confirms a growing burden of syphilis in the general population of Cameroon, and the need to reinforce surveillance and prevention strategies to fight STIs. To control syphilis infection, we need to involve keys actors like clinicians, because they must educate patients, counsel them in sexual risk reduction, and routinely screen those at increased risk. Finally, there is a need of a huge emphasizing of systematic screening of STIs particularly among people living with HIV. Strategies such as political advocacy of systematic counseling and testing of syphilis at ANC and treatment could be valuable. These strategies may target adult pregnant women, who are HIV-positive, single living in rural area priority in Southern Cameroon (South, East, Centre and North-West regions).

## Supporting information

**S1 Material.**
(DTA)

## Acknowledgments

We are grateful to pregnant women who provided consent to participate in this survey. We also thank the personnel of the PMTCT services and of the laboratories of all the participating health facilities. We thank all staff of National Aids Control Committee of Cameroon involved in training and data collection. We also thank the Ministry of Public Health for providing support for the implementation of the study throughout the country.

 **Disclaimer:** The views represented are those of the authors' alone and not their respective institutions or affiliated positions or funders.

## Author Contributions

**Conceptualization:** Cyprien Kengne-Nde, Jean de Dieu Anoubissi, Gabriel Loni-Ekali, Yasmine Moussa, Arlette Messeh, Joseph Fokam, Denis Snayeul-Wawo, Marinette Ngo-Nemb, David Kob, Serge-Clotaire Billong, Leonard Bonono, Jean-Bosco Elat.

**Data curation:** Cyprien Kengne-Nde, Jean de Dieu Anoubissi, Arlette Messeh, Joseph Fokam, Denis Snayeul-Wawo.

**Formal analysis:** Cyprien Kengne-Nde.

**Funding acquisition:** Serge-Clotaire Billong, Leonard Bonono, Jean-Bosco Elat.

**Investigation:** Cyprien Kengne-Nde, Jean de Dieu Anoubissi, Yasmine Moussa, Joseph Fokam, Albert Zeh-Meka, Denis Snayeul-Wawo, Dorine Tseuko, Marinette Ngo-Nemb, David Kob, Serge-Clotaire Billong, Leonard Bonono, Jean-Bosco Elat.

**Methodology:** Cyprien Kengne-Nde, Jean de Dieu Anoubissi, Gabriel Loni-Ekali, Arlette Messeh, Joseph Fokam, Denis Snayeul-Wawo, Dorine Tseuko, Marinette Ngo-Nemb, David Kob, Serge-Clotaire Billong, Leonard Bonono, Jean-Bosco Elat.

**Project administration:** Leonard Bonono, Jean-Bosco Elat.

**Resources:** Serge-Clotaire Billong, Leonard Bonono, Jean-Bosco Elat.

**Software:** Cyprien Kengne-Nde.

**Supervision:** Gabriel Loni-Ekali, Celine Nguefeu-Nkenfou, Yasmine Moussa, Arlette Messeh, Joseph Fokam, Albert Zeh-Meka, Denis Snayeul-Wawo, Dorine Tseuko, Marinette Ngo-Nemb, David Kob, Serge-Clotaire Billong, Jean-Bosco Elat.

**Validation:** Cyprien Kengne-Nde, Jean de Dieu Anoubissi, Gabriel Loni-Ekali, Celine Nguefeu-Nkenfou, Yasmine Moussa, Arlette Messeh, Joseph Fokam, Albert Zeh-Meka, Denis Snayeul-Wawo, Dorine Tseuko, Marinette Ngo-Nemb, David Kob, Serge-Clotaire Billong, Leonard Bonono, Jean-Bosco Elat.

**Visualization:** Cyprien Kengne-Nde, Gabriel Loni-Ekali, Yasmine Moussa, Denis Snayeul-Wawo, Leonard Bonono.

**Writing – original draft:** Cyprien Kengne-Nde.

**Writing – review & editing:** Cyprien Kengne-Nde, Jean de Dieu Anoubissi, Gabriel Loni-Ekali, Celine Nguefeu-Nkenfou, Yasmine Moussa, Arlette Messeh, Joseph Fokam, Albert Zeh-Meka, Denis Snayeul-Wawo, Dorine Tseuko, Marinette Ngo-Nemb, David Kob, Serge-Clotaire Billong, Leonard Bonono, Jean-Bosco Elat.

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
