## [Decision Letter · Decision Letter 0]

8 Jul 2020

PONE-D-20-11194

Highlighting a population-based re-emergence of Syphilis infection and assessing associated risk factors among pregnant women in Cameroon: Evidence from the 2009, 2012 and 2017 national sentinel surveillance surveys of HIV and syphilis.

PLOS ONE

Dear Dr. KENGNE-NDE,

Thank you for submitting your manuscript to PLOS ONE. After careful consideration, we feel that it has merit but does not fully meet PLOS ONE’s publication criteria as it currently stands. Therefore, we invite you to submit a revised version of the manuscript that addresses the points raised during the review process.

We look forward to receiving your revised manuscript.

Kind regards,

Remco PH Peters, MD, PhD, DLSHTM

Academic Editor

PLOS ONE

Journal Requirements:

- https://journals.plos.org/plosone/article?id=10.1371%2Fjournal.pone.0208963

- https://journals.sagepub.com/doi/10.1177/0956462415624058

In your revision ensure you cite all your sources (including your own works), and quote or rephrase any duplicated text outside the methods section. Further consideration is dependent on these concerns being addressed.

Reviewers' comments:

Reviewer's Responses to Questions

**Comments to the Author**

1. Is the manuscript technically sound, and do the data support the conclusions?

Reviewer #1: Partly

Reviewer #2: Partly

Reviewer #3: Partly

2. Has the statistical analysis been performed appropriately and rigorously? 

Reviewer #1: No

Reviewer #2: No

Reviewer #3: I Don't Know

3. Have the authors made all data underlying the findings in their manuscript fully available?

Reviewer #1: No

Reviewer #2: No

Reviewer #3: Yes

4. Is the manuscript presented in an intelligible fashion and written in standard English?

Reviewer #1: Yes

Reviewer #2: Yes

Reviewer #3: No

5. Review Comments to the Author

Reviewer #1: Thank you for the opportunity to review this manuscript. The manuscript a very important issue in public health and in the efforts towards the dual elimination of mother to child transmission of syphilis. The paper is generally well written but I think that the methods have not been sufficiently described or are inadequate to warrant the conclusions made. The authors conclude that there was a massive increase in syphilis prevalence among first ANC clinic attendees without providing sufficient evidence to exclude sampling issues, testing procedures, and errors in the analysis e.g. lack of weighting in the analysis. I have detailed my comments to the authors below

Abstract

• Line 3: This sentence should be syphilis and HIV can be transmitted from and not by

• Lines 8- 14: you need to clarify that the survey was conducted at the same sites every time and how many were enrolled in each survey.

• Lines 27-28: Please use word as suggest and not confirm as you haven’t provided evidence that syphilis has been re-emerging in Cameroonian general population besides the findings from the surveys

Introduction

• Line 33: sentence should read “Sub-Saharan African”. Please change this here and elsewhere where you have used this term – see line 54

• Line 42: remove “HIV and” and write sexual transmitted as sexually transmitted

• Line 67: in the methods, you refer to a nine year period and here to an eight. Please clarify

Methods

• Generally the methods need to be improved.

• Lines 70- 88: The design should be a secondary analysis of data from three ANC sentinel surveys. There after you need to put an extra section on the description of the three surveys. A table mighty be useful to highlight differences and similarities in the survey methods used across the 3 surveys Some important information to include is

o How often are the surveys done? Are they done on a regular basis or as needed?

o You say the surveys included 20 sentinel sites in 10 regions. Were these sentinel sites the same across all three surveys? How are the sentinel sites selected?

o You also say you included 10 surveillance health facilities. What is the nature of the surveillance health facilities and how do they differ or how are they related to the sentinel sites?

o What is the nature of a sentinel site – is it a clinic or hospital

• Lines 89- 110: This should be a sub-section of the proposed section on description of the surveys. Was syphilis testing conducted in all three surveys? If so how was it done in each if the survey? Later in the methods section (line 131- 134) you mentioned that only the 2017 did you have women tested for both HIV and syphilis at PMTCT sites. It’s not clear what this means? Where were the labs located in relation to the sentinel sites?

• Lines 124- 142: You did not define what your outcomes of the analysis are. Does the outcome have more than 2 responses? If yes which ones? This would justify use of the multinomial logistic regression.

• Since the data were collected in a survey where there might be clustering of outcomes, was the data weighted? Were the same data weights used across the 3 surveys? If no weights were applied, I would recommend a weighted analysis for calculating the survey specific syphilis sero-prevalence’s and even the analyses of trends across the surveys or at least some justification for why a weighted analysis is not necessary

Results

• Lines 151- 175: Please combine tables 1a, b and c to make it easier to compare the characteristics of participants across the surveys. Why not categorise age into fewer categories – 15- 24, 25- 35, 35- 49. Why were variable available different across the surveys?

• Lines 176- 183: Did you do a trend analysis? How? Please describe in the methods

• Lines 184: Figure 2 is not clear at all. A simple bar graph with error bars could be clearer

• Line 186- 192: what % of those enrolled in the 2017 survey did the 3901 tested for syphilis represent? Why were only these ones tested for syphilis? Please the information in the methods

• Line 196: please add the numbers for the total population and HIV positive population to table 2

• Also are the results in Table 2 weighted?

• Did you collect information on ART for the women who were HIV POSITIVE?

Discussion

• Line 248 – add HIV to co-infection

• Line 249- 253: I would tone down this assertion that syphilis prevalence is increasing given the methodological issues highlighted. Perhaps say maternal syphilis seroprevalence maybe increasing

• Have you looked at other data sources on maternal seropositivity e.g. from DHIS that you can triangulate with

• Line 259: not sure what middles in this sentence means

• Line 266- 269: you don’t discuss reasons why syphilis is higher in HIV positive women and how future surveys or studies can untangle this. The HIV/syphilis positive women – are they more likely to be on ART? Are they not on ART?

• Line 274: what was ANC attendance like in rural areas and has it changed over the 8/9 year period here? You don’t present rural urban distribution in the 2009 and 2012 surveys? It is possible that the increased prevalence is due to more rural women taking part in the survey?

Reviewer #2: This is an interesting study that provides a lot of food for thought: how can syphilis explode while HIV declines? Fascinating.

I have a few comments/suggestions, mostly minor but one major.

Minor

1. Did availability of antibiotics (e.g. over the counter purchases) change between the surveys?

2. line 57-59. This is stating the same thing twice. Also the causal link between syphilis and HIV is not realy proven, so please weaken your statement (e.g. syphilis has been implicated in susceptibility to HIV)

3. line 272. Please change "is the result" to "may be the result"

4. line 279. Is region associated with religion? If so please mention this.

5. While the ms is generally well written a few sentences seem to be not standard English. While acceptable it might be useful to have a native speaker check its grammar.

6. Were the same ANC sites used in the three surveys? This also impacts analysis

Major

1. Statistical analysis seems to ignore the structure/design of the survey which looks more like a multi-stage survey than a simple random sample. STATA offers excellent routines for analysing this type of data. Same applies to sample size calculation.

2. Perhaps the different logistic regression analyses can be applied to the all three surveys?

3. The increase in syphilis should also be demonstrated using logistic regression with survey year as one of the covariables. One can thereby adjust for changes in other risk factors.

Reviewer #3: The purpose of this article is to monitor changes in the seroprevalence of HIV/syphilis co-infection and syphilis infection and associated risk factors in Cameroon from 2009 to 2012 and 2017. These questions are important as they provide evidence for interventions for the prevention and control of HIV/AIDS and syphilis. To carry-out the objectives, the authors use cross-sectional antenatal care surveys conducted in 2009, 2012, and 2017 from 20 sentinel surveillance sites across 10 regions of Cameroon. The authors found the following:

1) HIV/syphilis co-infection increased from 0.05% in 2009 to 0.49% in 2017. Pregnant women aged 25–49 compared with those aged 15–24 were 15.1 times more likely to be co-infected. Single or unmarried compared to those who were married, cohabitating, widowed, or divorced were 2.9 times more likely to have a co-infection.

2) Syphilis infection increased from 0.6% in 2009 to 5.6% in 2017. Pregnant women living in rural areas were 1.8 times more likely to contract syphilis than those in urban areas. Pregnant women living in Northern Region were 0.6 times as likely to have a syphilis infection than those in the Southern Regions.

3) HIV infection decreased from 7.6% in 2009 to 5.7% in 2017. Pregnant women aged 25–49 compared with those aged 15–24 were 2.7 times more likely to be HIV positive. Multiparous pregnant women 2.0 times more likely to have an HIV infection than non-nulliparous.

This work provides evidence for national policy, resource allocation, and the evidence base for interventions for the control and prevention of HIV and syphilis in Cameroon and the global elimination of mother-to-child transmission of HIV and syphilis.

Comments/Questions:

1. In the methods section, please specify the sampling method used. Specify probability or nonprobability sampling and type of sampling (ie. simple random, systematic, convenience, …) that was used.

2. Did the analysis have 2 or more dependent or outcome variables or were there multiple independent or response variables? The methods mention multivariate and multivariable interchangeably.

3. The results sections is bit hard to follow. Consider grouping results by the specific infection and co-infection so that the reader can look to the results to inform the story that is being told in the discussion section.

4. On line 156, what is meant by the term ‘monogomous regime’?

5. “Enroled’ (British English) and ‘enrolled’ (American English) are used interchangeably throughout the paper. Choose one or the other according to the journal’s style guide.

6. Also check for capitalization where it is not needed (ie. see line 270 “In contrast of Syphilis trend”, line 279, …).

6. PLOS authors have the option to publish the peer review history of their article (what does this mean?). If published, this will include your full peer review and any attached files.

Reviewer #1: **Yes: **Tendesayi Kufa

Reviewer #2: **Yes: **Nico Nagelkerke

Reviewer #3: No

---

## [Author Response · Author response to Decision Letter 0]

18 Sep 2020

Editor

Authors: Our manuscript meets PLOS ONE's style requirements, including those for file naming, Thank you.

- https://journals.plos.org/plosone/article?id=10.1371%2Fjournal.pone.0208963

- https://journals.sagepub.com/doi/10.1177/0956462415624058

In your revision ensure you cite all your sources (including your own works), and quote or rephrase any duplicated text outside the methods section. Further consideration is dependent on these concerns being addressed.

Authors: In our revision we cite or rephrase duplicated text outside the methods section. Thank you.

Authors: According to the national survey data sharing policies dictated by the ethic committee, access to data of 2009 and 2012 data survey is not systematic and is subject to a request via infos@cnls.cm; Phone: (00237) 222 22 57 58, Fax: 222 23 34 39; P.O BOX: 1459 Yaounde, Cameroon. Nevertheless, data for 2017 survey is not submitted to this restriction and we submitted it with this revised manuscript. Thank you.

4. Please amend your list of authors on the manuscript to ensure that each author is linked to an affiliation. Authors’ affiliations should reflect the institution where the work was done (if authors moved subsequently, you can also list the new affiliation stating “current affiliation:.” as necessary).

Authors: The list of authors has been amended. Each author is linked to an affiliation. Thank you.

Reviewer #1:

Thank you for the opportunity to review this manuscript. The manuscript a very important issue in public health and in the efforts towards the dual elimination of mother to child transmission of syphilis. The paper is generally well written but I think that the methods have not been sufficiently described or are inadequate to warrant the conclusions made. The authors conclude that there was a massive increase in syphilis prevalence among first ANC clinic attendees without providing sufficient evidence to exclude sampling issues, testing procedures, and errors in the analysis e.g. lack of weighting in the analysis. I have detailed my comments to the authors below.

Authors: We thank the Reviewer 1 for this summary.

Abstract

• Line 3: This sentence should be syphilis and HIV can be transmitted from and not by

Authors: Thank you for this comment. We modified the sentence in the manuscript.

• Lines 8- 14: you need to clarify that the survey was conducted at the same sites every time and how many were enrolled in each survey.

Authors: Thank you for this comment. We clarified in the manuscript.

• Lines 27-28: Please use word as suggest and not confirm as you haven’t provided evidence that syphilis has been re-emerging in Cameroonian general population besides the findings from the surveys.

Authors: Thank you for this comment. We used the word “suggest” in the manuscript.

Introduction

• Line 33: sentence should read “Sub-Saharan African”. Please change this here and elsewhere where you have used this term – see line 54

Authors: Thank you for this comment. We changed the term “Sub-Saharan” with “Sub-Saharan African” here and elsewhere we have used it in the manuscript.

• Line 42: remove “HIV and” and write sexual transmitted as sexually transmitted

Authors: Thank you for this comment. We modified the sentence in the manuscript.

• Line 67: in the methods, you refer to a nine-year period and here to an eight. Please clarify

Authors: Thank you for this remark. We corrected the error to a “nine-year period” in the manuscript.

Methods

• Generally, the methods need to be improved.

• Lines 70- 88: The design should be a secondary analysis of data from three ANC sentinel surveys. There after you need to put an extra section on the description of the three surveys. A table mighty be useful to highlight differences and similarities in the survey methods used across the 3 surveys Some important information to include is:

Authors: We thank reviewer 1 for this suggestion. Since the surveys were very similar across the year, we specified details in the paragraph of the “Description of the surveys, study design and participants” section. Thank you.

o How often are the surveys done? Are they done on a regular basis or as needed?

Authors: The survey are supposed to be done in a Regular basis (2 years). But, according to funds availability, one survey for a specific year could be postponed. Thank you.

o You say the surveys included 20 sentinel sites in 10 regions. Were these sentinel sites the same across all three surveys? How are the sentinel sites selected?

Authors: Thank you for this important question which can help others authors in the reproductibility of this study. The sentinel sites were the same across all the three surveys. As already described in the manuscript (Line 76 – 79), The sentinel sites were chosen based on their capacity to provide both ANC and PMTCT services, their location (urban and rural settings in each region of the country), ANC attendance (capacity to enrol at least 300 pregnant ANC1 attendees during the study period of three months).

o You also say you included 10 surveillance health facilities. What is the nature of the surveillance health facilities and how do they differ or how are they related to the sentinel sites?

Authors: Thank you for this question. As described in the manuscript (line 73-76), we conducted the surveys in 20 sentinel sites that included 60 HIV surveillance health facilities (routine collection points). So, each sentinel site has 3 HIV surveillance health facilities. The surveillance health facilities were health facilities from public and private sector. The collection point was found in a sentinel site (related to a specific geographic area of the region). These collection points are hospitals with high ANC attendance that meet the selection criteria as recommended by the WHO.

o What is the nature of a sentinel site – is it a clinic or hospital

Authors: A sentinel site is a specific geographic area in the region. In the sentinel site, HIV surveillance health facilities (clinic and or hospital) were the data collection points. Each sentinel site had 3 data collection points. Thank you.

• Lines 89- 110: This should be a sub-section of the proposed section on description of the surveys. Was syphilis testing conducted in all three surveys? If so how was it done in each if the survey? 

Authors: Yes, the syphilis testing was conducting in all three surveys. It was done using the Treponema Palladium Hemagglutination assay (TPHA) / Veneral Diseases Research Laboratory (VDRL) as per the manufacturer’s instructions and WHO recommendations. Thank you.

Later in the methods section (line 131- 134) you mentioned that only the 2017 did you have women tested for both HIV and syphilis at PMTCT sites. It’s not clear what this means? 

Authors: HIV testing was conducted for women at the sites and at the reference laboratory. Results from the reference laboratory were used. Be it 2009, in 2012, Syphilis test was performed consistently at the level of the PMTCT sites using the Treponema Palladium Hemagglutination assay (TPHA) / Veneral Diseases Research Laboratory (VDRL) and results were synthesized at the national level. Thank you.

Where were the labs located in relation to the sentinel sites?

Authors: The labs were found in each of the PMTCT sites and this was a condition to select that site as a surveillance health facility. The reference lab was found in the capital of the country (Yaounde) and tested all the samples collected from sentinel sites. Thank you.

• Lines 124- 142: You did not define what your outcomes of the analysis are. Does the outcome have more than 2 responses? If yes which ones? This would justify use of the multinomial logistic regression.

Authors: The main outcome of our analysis was HIV/syphilis co-infection variable which has 4 responses: 

Response 1 = HIV/syphilis co-infection 

Response 2 = HIV infection only 

Response 3 = Syphilis infection only 

Response 4 = No infection

We specify this in the method section (Line 141).

Thank you.

• Since the data were collected in a survey where there might be clustering of outcomes, was the data weighted? Were the same data weights used across the 3 surveys? If no weights were applied, I would recommend a weighted analysis for calculating the survey specific syphilis sero-prevalence’s and even the analyses of trends across the surveys or at least some justification for why a weighted analysis is not necessary.

Authors: Based on the WHO protocol for HIV Sentinel Surveillance Survey (Reference 17 in the manuscript) we have implemented a non-probabilistic approach which consisted of a systematic sampling method with two stages

(1) Selection of sentinel sites / surveillance health facilities (routine collection points)

(2) Selection of pregnant women in each study sentinel site. Briefly, study sites were selected by based on: (a) representativeness at regional level, (b) geographical location in each region (urban or rural), (c) availability of PMTCT services and data management system, (d) functionality of the site (staff and materials for ANC/PMTCT, laboratory services and a cold chain), and (e) site willingness for participation into the study. Selected sites were from health facilities of the primary, secondary and tertiary healthcare levels. 

However, since the sample is not representative of the population, we think that it is not necessary to have a weighted analysis approach since the objective of WHO sentinel studies is to have a trend of the epidemic and not to extrapolate the results.

Please feel free to find more details in the reference below [1]:

[1] « WHO | Guidelines for conducting HIV sentinel serosurveys among pregnant women and other groups », WHO, mars 21, 2020. https://www.who.int/hiv/pub/surveillance/anc_guidelines/en/ (consulté le mars 21, 2020).

 Thank you.

Results

• Lines 151- 175: Please combine tables 1a, b and c to make it easier to compare the characteristics of participants across the surveys. 

Authors: The tables 1a, b and c have been combined into one (table 1 line169), Thank you.

Why not categorise age into fewer categories – 15- 24, 25- 35, 35- 49. 

Authors: We harmonize the categories of age in the three survey. We considered 15-19 and 20-24 to separate adolescent (15-19) and teenager (20-24), Thank you.

Why were variable available different across the surveys?

Authors: The variable available were different across the surveys because the questionnaire have slightly changed from one survey to another according to the UNAIDS, WHO and the Country strategies and priority. Thank you.

• Lines 176- 183: Did you do a trend analysis? How? Please describe in the methods

Authors: No, we did not do a trend analysis, but we estimated according to the WHO protocol each survey’s seroprevalence and the Confidence Interval that allowed the comparisons to be made. Thank you.

• Lines 184: Figure 2 is not clear at all. A simple bar graph with error bars could be clearer

Authors: We have tried to improve the quality of this graphic and we plead the reviewer 1 to accept the current version. Thank you.

• Line 186- 192: what % of those enrolled in the 2017 survey did the 3901 tested for syphilis represent? 

Authors: The 3901 tested for syphilis represented 56.9% of those enrolled in the 2017 survey. This has been had in the manuscript. Thank you.

Why were only these ones tested for syphilis? Please the information in the methods

Authors: Thank you for this important question. Other pregnant women were tested but those with inconclusive results were not considered in the analysis. Those who could not benefit from syphilis testing due to stock out were also excluded from the analysis. The information has been added in the methods section already. 

• Line 196: please add the numbers for the total population and HIV positive population to table 2

Authors: We added the numbers for the total population and HIV positive population to table 2. Thank you.

• Also are the results in Table 2 weighted?

Authors: No, the results in Table 2 are not weighted according to the non-probabilistic approach implemented. Thank you.

• Did you collect information on ART for the women who were HIV POSITIVE?

Authors: No, we did not collect this information since it is not included in the WHO protocol.

This could be used to improve the WHO protocol on this aspect in the area of Test and Treat

We use this contribution to improve the discussion part on this aspect. Thank you.

Discussion

• Line 248 – add HIV to co-infection

Authors: We added HIV to co-infection. Thank you.

• Line 249- 253: I would tone down this assertion that syphilis prevalence is increasing given the methodological issues highlighted. Perhaps say maternal syphilis seroprevalence maybe increasing

Authors: Thank you to the reviewer for this. We modified the sentence accordingly. 

• Have you looked at other data sources on maternal seropositivity e.g. from DHIS that you can triangulate with

Authors: No, we did not triangulate with DHIS data because the country adopted DHIS recently (2015) and is not collecting Syphilis data routinely as from now. But for HIV we observed the decreases of Seropositivity like in Sentinels surveillance surveys. DHIS is not used now to collect all data (Syphilis, hepatitis, etc.). Nevertheless, it could be really interesting to use these data if collected routinely trough DHIS to monitor and triangulate data of these disease in the future if we include this to DHIS and reduce cost of these survey. We use this contribution to improve the discussion part on this aspect. Thank you.

• Line 259: not sure what middles in this sentence means

Authors: Middles was used to range the seroprevalence found in the country: not small and not high comparing to the ones found in other Sub-Saharan African countries. We modified the sentence to make it clear. Thank you.

• Line 266- 269: you don’t discuss reasons why syphilis is higher in HIV positive women and how future surveys or studies can untangle this. The HIV/syphilis positive women – are they more likely to be on ART? Are they not on ART?

Authors: Thank you to reviewer 1 for this interesting question. Unfortunately, we did not collect information about ART. Nevertheless, pregnant women living with HIH may have a weaker immune system. We use this contribution to improve the discussion section on this point. Thank you.

• Line 274: what was ANC attendance like in rural areas and has it changed over the 8/9 year period here? You don’t present rural urban distribution in the 2009 and 2012 surveys? It is possible that the increased prevalence is due to more rural women taking part in the survey?

Authors: We added rural urban distribution in the 2009 and 2012 surveys in Table 1 (Line 170), and about two fifth of women enrolled were living in urban area in all the three surveys (40% in 2009; 42.7% in 2012 and 42.7% in 2017). We can thus conclude that the increased prevalence is not due to more rural women taking part in the survey. Thank you.

Reviewer #2: 

This is an interesting study that provides a lot of food for thought: how can syphilis explode while HIV declines? Fascinating. 

Authors: We thank the Reviewer 2 for this appreciation.

I have a few comments/suggestions, mostly minor but one major.

Minor

1. Did availability of antibiotics (e.g. over the counter purchases) change between the surveys?

Authors: No, availability of antibiotics did not change between the surveys. Thank you.

2. line 57-59. This is stating the same thing twice. Also the causal link between syphilis and HIV is not realy proven, so please weaken your statement (e.g. syphilis has been implicated in susceptibility to HIV)

Authors: We thank the reviewer 2 for this remark. We modified our statement accordingly.

3. line 272. Please change "is the result" to "may be the result"

Authors: We thank the reviewer 2 for this comment. We modified our statement accordingly.

4. line 279. Is region associated with religion? If so, please mention this.

Authors: Yes, in the country (Cameroon) Northern regions are most muslim and southern ones are more Christian. We mention this in the discussion section. Thank you.

5. While the ms is generally well written a few sentences seem to be not standard English. While acceptable it might be useful to have a native speaker check its grammar.

Authors: The suggestion have been considered. Thank you.

6. Were the same ANC sites used in the three surveys? This also impacts analysis

Authors: Yes, the sentinel surveillance sites were the same in the three surveys, Thank you.

Major

1. Statistical analysis seems to ignore the structure/design of the survey which looks more like a multi-stage survey than a simple random sample. STATA offers excellent routines for analysing this type of data. Same applies to sample size calculation.

Authors: Based on the WHO protocol for HIV Sentinel Surveillance Survey (Reference 17 in the manuscript) we have implemented a non-probabilistic approach which consisted of a systematic sampling method with two stages

(1) Selection of sentinel sites / surveillance health facilities (routine collection points)

(2) Selection of pregnant women in each study sentinel site. Briefly, study sites were selected by based on: (a) representativeness at regional level, (b) geographical location in each region (urban or rural), (c) availability of PMTCT services and data management system, (d) functionality of the site (staff and materials for ANC/PMTCT, laboratory services and a cold chain), and (e) site willingness for participation into the study. Selected sites were from health facilities of the primary, secondary and tertiary healthcare levels. 

However, since the sample is not representative of the population, we think that it is not necessary to have a weighted analysis approach since the objective of WHO sentinel studies is to have a trend of the epidemic and not to extrapolate the results.

Please feel free to find more details in the reference below [1]:

[1] « WHO | Guidelines for conducting HIV sentinel serosurveys among pregnant women and other groups », WHO, mars 21, 2020. https://www.who.int/hiv/pub/surveillance/anc_guidelines/en/ (consulté le mars 21, 2020).

 Thank you.

2. Perhaps the different logistic regression analyses can be applied to the all three surveys?

Authors: We limited our analysis to logistic regression in 2017 to have more recent associative factors to guide decision making and strategies more efficiently. Thank you.

3. The increase in syphilis should also be demonstrated using logistic regression with survey year as one of the covariables. One can thereby adjust for changes in other risk factors.

Authors: Thank you for this suggestion. But we think that it could not be right to use the survey year using our logistic regression since we use only the 2017 data survey in the model. It could have been interesting to do so if we did a generalized logistic regression model (with mixed effect or GEE approach), but even in this case it will also challenging to implement since the participants were not the same from one-year study to another. Thank you.

Reviewer #3: 

The purpose of this article is to monitor changes in the seroprevalence of HIV/syphilis co-infection and syphilis infection and associated risk factors in Cameroon from 2009 to 2012 and 2017. These questions are important as they provide evidence for interventions for the prevention and control of HIV/AIDS and syphilis. To carry-out the objectives, the authors use cross-sectional antenatal care surveys conducted in 2009, 2012, and 2017 from 20 sentinel surveillance sites across 10 regions of Cameroon. The authors found the following:

1) HIV/syphilis co-infection increased from 0.05% in 2009 to 0.49% in 2017. Pregnant women aged 25–49 compared with those aged 15–24 were 15.1 times more likely to be co-infected. Single or unmarried compared to those who were married, cohabitating, widowed, or divorced were 2.9 times more likely to have a co-infection.

2) Syphilis infection increased from 0.6% in 2009 to 5.6% in 2017. Pregnant women living in rural areas were 1.8 times more likely to contract syphilis than those in urban areas. Pregnant women living in Northern Region were 0.6 times as likely to have a syphilis infection than those in the Southern Regions.

3) HIV infection decreased from 7.6% in 2009 to 5.7% in 2017. Pregnant women aged 25–49 compared with those aged 15–24 were 2.7 times more likely to be HIV positive. Multiparous pregnant women 2.0 times more likely to have an HIV infection than non-nulliparous.

This work provides evidence for national policy, resource allocation, and the evidence base for interventions for the control and prevention of HIV and syphilis in Cameroon and the global elimination of mother-to-child transmission of HIV and syphilis.

Authors: We really appreciate the summary did by the reviewer 3. Thank you.

Comments/Questions:

1. In the methods section, please specify the sampling method used. Specify probability or nonprobability sampling and type of sampling (ie. simple random, systematic, convenience, …) that was used.

Authors: Based on the WHO protocol for HIV Sentinel Surveillance Survey (Reference 17 in the manuscript) we have implemented a non-probabilistic approach which consisted of a systematic sampling method with two stages

(1) Selection of sentinel sites / surveillance health facilities (routine collection points)

(2) Selection of pregnant women in each study sentinel site. Briefly, study sites were selected by based on: (a) representativeness at regional level, (b) geographical location in each region (urban or rural), (c) availability of PMTCT services and data management system, (d) functionality of the site (staff and materials for ANC/PMTCT, laboratory services and a cold chain), and (e) site willingness for participation into the study. Selected sites were from health facilities of the primary, secondary and tertiary healthcare levels. 

We specify this in the method section.

Thank you.

2. Did the analysis have 2 or more dependent or outcome variables or were there multiple independent or response variables? The methods mention multivariate and multivariable interchangeably.

Authors: The dependent variable of our analysis was HIV/syphilis co-infection which has 4 responses: 

Response 1 = HIV/syphilis co-infection 

Response 2 = HIV infection only 

Response 3 = Syphilis infection only 

Response 4 = No infection

We specify this in the method section.

Thank you.

3. The results sections is bit hard to follow. Consider grouping results by the specific infection and co-infection so that the reader can look to the results to inform the story that is being told in the discussion section.

Authors: We thank the reviewer 3 for this suggestion. We combined Table 1a, b and C into one table to make it easier to the reader. Thank you.

4. On line 156, what is meant by the term ‘monogomous regime’?

Authors: “Monogomous regime” means to be married with one partner. This have been remove to harmonise with the results of other years. Thank you.

5. “Enroled’ (British English) and ‘enrolled’ (American English) are used interchangeably throughout the paper. Choose one or the other according to the journal’s style guide.

Authors: We harmonized and use enrolled (American English). Thank you.

6. Also check for capitalization where it is not needed (ie. see line 270 “In contrast of Syphilis trend”, line 279, …).

Authors: We checked for capitalization where it is not needed, Thank you.

---

## [Decision Letter · Decision Letter 1]

9 Oct 2020

PONE-D-20-11194R1

Highlighting a population-based re-emergence of Syphilis infection and assessing associated risk factors among pregnant women in Cameroon: Evidence from the 2009, 2012 and 2017 national sentinel surveillance surveys of HIV and syphilis.

PLOS ONE

Dear Dr. KENGNE-NDE,

Thank you for submitting your manuscript to PLOS ONE. After careful consideration, we feel that it has merit but does not fully meet PLOS ONE’s publication criteria as it currently stands. Therefore, we invite you to submit a revised version of the manuscript that addresses the points raised during the review process.

I would be happy to accept your manuscript for publication once you have addressed the reviewer's minor comments.

We look forward to receiving your revised manuscript.

Kind regards,

Remco PH Peters, MD, PhD, DLSHTM

Academic Editor

PLOS ONE

Reviewers' comments:

Reviewer's Responses to Questions

**Comments to the Author**

1. If the authors have adequately addressed your comments raised in a previous round of review and you feel that this manuscript is now acceptable for publication, you may indicate that here to bypass the “Comments to the Author” section, enter your conflict of interest statement in the “Confidential to Editor” section, and submit your "Accept" recommendation.

Reviewer #1: (No Response)

Reviewer #2: All comments have been addressed

2. Is the manuscript technically sound, and do the data support the conclusions?

Reviewer #1: Partly

Reviewer #2: Yes

3. Has the statistical analysis been performed appropriately and rigorously? 

Reviewer #1: No

Reviewer #2: Yes

4. Have the authors made all data underlying the findings in their manuscript fully available?

Reviewer #1: Yes

Reviewer #2: No

5. Is the manuscript presented in an intelligible fashion and written in standard English?

Reviewer #1: Yes

Reviewer #2: Yes

6. Review Comments to the Author

Reviewer #1: Thank you for the opportunity to review this revised manuscript. The authors have addressed most of the comments I had on initial review. The paper reads much better than the initial version and the methods section clearer. However in addressing these, new comments have arisen. I have listed them below:

Abstract

Line 9: add each year

Line 13: add outcomes of the multinomial logistic regression

Introduction

Line 47 – the authors need to add the phrase “such as ” before HIV and syphilis

Line 53- delete the words the highest risk. It sounds repetitive with most in that sentence

Lines 60- 63: the authors don’t address why syphilis is an important co-infection in HIV + patients. Does it have an effect on HIV progression

Methods

Lines 73- 89: The authors clarified what constituted a sentinel site – health facility including its 3 testing points. This is not clear in clear in this paragraph

Lines 112- 115 syphilis testing – how was a case of syphilis decided. Please include the algorithm like you have for HIV. Was the algorithm reverse or traditional. Were titres measured for VDRL to determine current vs old infection

Lines 130- Is the CSPRO software a registered brand? Add developer, city and country

Line 138 – the authors refer to a sub-set to be tested for syphilis. How was this subset selected. Later on they state that shortage of test kits meant sites with no test kits could not be selected for the analysis.

Results

Line 163 – please change from involvement to particioation

Table 1- present medians (IQR) as well as age in cvategories

Table 2 – present numbers before percentages

Table 3: for overall population and those HIV positive add the values of N- the totals

Also primiparous and number of pregnancies overlap as variables. Primiparous and # pregnancies =0 are one and the same thing.

Table 4: see comment above

Fig 1: add syphilis testing algorithm

Discussion

Lines 278- 286: the authors have not addressed the issue of access to treatment as a factor that could explain high rates of syphilis in rural areas. From about 2016, there has been a global benzathine penicillin shortage. Please comment on this

Lines 296- 308: The list of limitations is incomplete. Because selection of sites for syphilis testing were not random, discuss that it was possible that sites which higher syphilis rates could have been included in the survey

Reviewer #2: No further concerns. xxxxxxxxxxxxxxxxxxxxxxxxxxxxxxxxxxxxxxxxxxxxxxxxxxxxxxxxxxxxxxxxxxxxxxxxxxxxxxxxxxxxx

7. PLOS authors have the option to publish the peer review history of their article (what does this mean?). If published, this will include your full peer review and any attached files.

Reviewer #1: **Yes: **Tendesayi Kufa

Reviewer #2: **Yes: **Nico Nagelkerke

---

## [Author Response · Author response to Decision Letter 1]

22 Oct 2020

Reviewer #1:

Thank you for the opportunity to review this revised manuscript. The authors have addressed most of the comments I had on initial review. The paper reads much better than the initial version and the methods section clearer. However, in addressing these, new comments have arisen. I have listed them below:

Authors: We thank the Reviewer 1 for this appreciation.

Abstract

• Line 9: add each year 

Authors: We added each year in the sentence in the manuscript, Thank you.

• Line 13: add outcomes of the multinomial logistic regression

Authors: We added outcomes of the multinomial logistic regression in the manuscript, Thank you.

Introduction

• Line 47 – the authors need to add the phrase “such as” before HIV and syphilis 

Authors: We added “such as” before HIV and syphilis in the manuscript. Thank you.

• Line 53- delete the words the highest risk. It sounds repetitive with most in that sentence

Authors: The words the highest have been deleted. Thank you.

• Lines 60- 63: the authors don’t address why syphilis is an important co-infection in HIV + patients. Does it have an effect on HIV progression

Authors: Thank you to reviewer 1 for this important comment. We have modified the sentence by changing “Co-infection” with “risk factor” since there is no scientific evidence according to our knowledge on the direct influence of syphilis on the progression of HIV.

Methods

• Lines 73- 89: The authors clarified what constituted a sentinel site – health facility including its 3 testing points. This is not clear in clear in this paragraph

• Authors: We clarified more in the manuscript. Thank you. 

• Lines 112- 115 syphilis testing – how was a case of syphilis decided. Please include the algorithm like you have for HIV. Was the algorithm reverse or traditional. Were titres measured for VDRL to determine current vs old infection

Authors: Thank you to reviewer for this comment. The syphilis case was decided as indicated on the syphilis screening algorithm included as Figure 1.b. Yes, titres were measured for VDRL for the majority of the sentinel sites to determine if it was a current infection or an old/scarring infection, but unfortunately this information was not collected during the survey. We also add this as a limitation in the discussion. The algorithm was traditional.

• Lines 130- Is the CSPRO software a registered brand? Add developer, city and country

Authors: We have added developer, city and country, Thank you.

• Line 138 – the authors refer to a sub-set to be tested for syphilis. How was this subset selected. Later on they state that shortage of test kits meant sites with no test kits could not be selected for the analysis.

Authors: Thank you to reviewer for this comment. The subset of analysis was selected based on the available syphilis test results. In fact, all sites performed syphilis testing, but the stockout experienced by some sites at one point during the study unfortunately led to some study participants not being tested and were excluded from the analysis. 

Results

• Line 163 – please change from involvement to particioation

Authors: We changed involvement to participation, Thank you.

• Table 1- present medians (IQR) as well as age in cvategories

Authors: We have prevented medians (IQR) for age and we have commented, Thank you.

• Table 2 – present numbers before percentages 

Authors: We have presented numbers before percentages Thank you.

• Table 3: for overall population and those HIV positives add the values of N- the totals. Also primiparous and number of pregnancies overlap as variables. Primiparous and # pregnancies =0 are one and the same thing.

Authors: Thank you for this interesting comment. The totals for the overall population and the HIV-positive were added. For the variables Primiparous and number of pregnancies, they were kept in the analysis to test in parallel the effect of first pregnancy on the risk of Syphilis and HIV infection and to see if a particular effect was found in the heterogeneous group of multiparous women in the risk of Syphilis and HIV infection via the variable number of pregnancies.

• Table 4: see comment above

Authors: Thank you for this pertinent comment. For the variables Primiparous and number of pregnancies, they were kept in the analysis to test in parallel the effect of first pregnancy on the risk of Syphilis and HIV infection and to see if a particular effect was found in the heterogeneous group of multiparous women via the variable number of pregnancies. Because these variables overlap, we considered only the primiparous variable in our multivariate analysis.

• Fig 1: add syphilis testing algorithm

Authors: The Syphilis algorithm have been added as Figure 1.b and Fig 1 is now Fig 1.a, Thank you.

Discussion

• Lines 278- 286: the authors have not addressed the issue of access to treatment as a factor that could explain high rates of syphilis in rural areas. From about 2016, there has been a global benzathine penicillin shortage. Please comment on this

Authors: Thank you to reviewer 1 for this interesting comment. As recommended in the WHO guide, syphilis (new or old infection) is cure in first intention with benzathine penicillin. WHO also recommended when benzathine or procaine penicillin cannot be used (e.g. due to penicillin allergy where penicillin desensitization is not possible) or are not available (e.g. due to stockouts), the WHO STI guideline suggests using, with caution, erythromycin 500 mg orally four times daily for 30 days [1]. Although erythromycin treats the pregnant women, it does not cross the placental barrier completely and as a result the fetus is not treated. It is therefore necessary to treat the newborn infant soon after delivery (see recommendations 9 and 10 in the WHO guidelines for the treatment of syphilis, which refer to congenital syphilis). Doxycycline should not be used in pregnant women. Because syphilis during pregnancy can lead to severe adverse complications to the fetus or newborn, stockouts of benzathine penicillin for use in antenatal care should be avoided [1].

Thus the global stockout of benzathine penicillin in 2016 could have play a role in the sero-prevalence rate and the spread of syphilis, particularly in rural areas[2]. We added this in the discussion (lines 291- 294).

Reference: [1] « WHO | WHO guideline on syphilis screening and treatment for pregnant women », WHO. http://www.who.int/reproductivehealth/publications/rtis/syphilis-ANC-screenandtreat-guidelines/en/ (consulté le oct. 11, 2020).

[2] S. Nurse-Findlay et al., « Shortages of benzathine penicillin for prevention of mother-to-child transmission of syphilis: An evaluation from multi-country surveys and stakeholder interviews », PLoS Med, vol. 14, no 12, déc. 2017, doi: 10.1371/journal.pmed.1002473.

Lines 296- 308: The list of limitations is incomplete. Because selection of sites for syphilis testing were not random, discuss that it was possible that sites which higher syphilis rates could have been included in the survey

Authors: Thank you to reviewer for this comment. We had included this limitation in the actual version of the manuscript (line 291 – 294).

---

## [Editor Report · Decision Letter 2]

26 Oct 2020

Highlighting a population-based re-emergence of Syphilis infection and assessing associated risk factors among pregnant women in Cameroon: Evidence from the 2009, 2012 and 2017 national sentinel surveillance surveys of HIV and syphilis.

PONE-D-20-11194R2

Dear Dr. KENGNE-NDE,

We’re pleased to inform you that your manuscript has been judged scientifically suitable for publication and will be formally accepted for publication once it meets all outstanding technical requirements.

Kind regards,

Remco PH Peters, MD, PhD, DLSHTM

Academic Editor

PLOS ONE